# Methodological choices and clinical usefulness for machine learning predictions of outcome in Internet-based cognitive behavioural therapy

Nils Hentati Isacsson [1] ✉, Fehmi Ben Abdesslem[1,2,3], Erik Forsell [1], Magnus Boman [3,4] & Viktor Kaldo[1,5]

## Abstract

**Background** While psychological treatments are effective, a substantial portion of patients do not benefit enough. Early identification of those may allow for adaptive treatment strategies and improved outcomes. We aimed to evaluate the clinical usefulness of machine-learning (ML) models predicting outcomes in Internet-based Cognitive Behavioural Therapy, to compare ML-related methodological choices, and guide future use of these.

**Methods** Eighty main models were compared. Baseline variables, weekly symptoms, and treatment activity were used to predict treatment outcomes in a dataset of 6695 patients from regular care.

**Results** We show that the best models use handpicked predictors and impute missing data. No ML algorithm shows clear superiority. They have a mean balanced accuracy of 78.1% at treatment week four, closely matched by regression (77.8%).

**Conclusions** ML surpasses the benchmark for clinical usefulness (67%). Advanced and simple models perform equally, indicating a need for more data or smarter methodological designs to confirm advantages of ML.

## Plain language summary

While there are many therapy treatments that are effective for mental health problems some patients don't benefit enough. Predicting whom might need more help can guide therapists to adjust treatments for better results. Computer methods are increasingly used for predicting the outcome of treatment, but studies vary widely in accuracy and methodology. We examined a variety of models to test performance. Those examined were based on a several factors: what data is chosen, how the data is managed, as well as type of mathematical equations and function used for prediction. When used on ~6500 patients, none of the computer methods tested stood out as the best. Simple models were as accurate as more advanced. Accuracy of prediction of treatment outcome was good enough to inform clinicians' decisions, suggesting they may still be useful tools in mental health care.

Evidence-based psychological treatments are beneficial for many conditions, and fewer treatments fail if therapists are given predictions for each patient's outcome, based on continuous monitoring of symptoms[1,2]. This enables the therapist to adjust treatment for patients risking failure, and is referred to as an Adaptive Treatment Strategy[2]. This has shown to improve outcomes in traditional psychological treatments[1,3].

Internet-delivered Cognitive Behavioural Therapy (ICBT), i.e. digital, diagnosis-specific self-help material with brief therapist support, could help

increase access to psychological treatment and shows effects similar to traditional face-to-face CBT[4]. This guided ICBT treatment format is extensively used in regular care worldwide[5] as well as thoroughly researched[6]. As in traditional CBT, about 30–60% of ICBT patients do not benefit sufficiently[7], but the use of an Adaptive Treatment Strategy has been shown to improve outcome[2].

Various approaches to predicting patients' treatment outcomes have been used, and the use of machine learning, supposedly superior to

[1]Centre for Psychiatry Research, Department of Clinical Neuroscience, Karolinska Institutet, & Stockholm Health Care Services, Region Stockholm, Sweden.
[2]Department of Computer Science, RISE Research Institutes of Sweden, Stockholm, Sweden. [3]Division of Psychiatry, University College London, London, UK.
[4]Department of Medicine Solna, Clinical Epidemiology Division, Karolinska Institutet, Stockholm, Sweden. [5]Department of Psychology, Faculty of Health and Life Sciences, Linnaeus University, Växjö, Sweden. ✉e-mail: nils.isacsson@ki.se

regression models when trained on large samples with multiple predictors[8], is growing within mental health care[9]. In both guided ICBT and traditional psychotherapy, using only baseline data as predictors results in weak predictions[10,11]. However, it is well established that early symptom change during treatment is associated with the symptomatic treatment outcome[12,13] and including predictors from the first weeks of treatment, as in Routine Outcome Monitoring[14] and Adaptive Treatment Strategies, increases the accuracy of predictions compared to baseline predictors only[15,16]. While early symptom change is fundamental as predictor, adding other predictors in conjunction with machine learning could however potentially increase the accuracy further. Bennemann and colleagues[17] identified that algorithms incorporating predictor selection and selecting from several variables, such as tree-based and boosted machine learning algorithms are beneficial for accurately predicting therapy dropout, an outcome distinct from symptomatic outcome yet likely linked.

The level of accuracy needed for a model to be clinically useful can vary, but[18] showed that an accuracy of 65% was deemed clinically actionable by clinicians, and Forsell with colleagues[19] showed that 67% accuracy in predicting symptom-related treatment outcome was enough to be clinically useful for therapists using an adaptive treatment strategy in a randomized trial.

Past studies vary widely in both the accuracy achieved and methodology used, and while early change has a large effect on treatment outcome[12], it is difficult to ascertain if, or at what week in treatment, prediction accuracy is good enough to be used in an Adaptive Treatment Strategy. Additionally, previous machine learning efforts often omit relevant comparisons to alternative prediction models and/or benchmarks[20] like those mentioned above and suffer from limited sample sizes[9,21]. In these studies, larger samples and thorough model-building and validation procedures often show lower predictive accuracy, stressing the need for rigorous methodological practices to avoid inflated results. Studies also vary in what predictors are used and how they are selected, strategies to deal with missing data, what predictive algorithms are used, and how (or if) the models are validated to avoid exaggerated accuracy due to overfitting. In general time-series forecasting studies, statistical and machine learning methods have shown different levels of accuracy[22], with a higher accuracy for statistical methods: however, the results' applicability to prediction of psychological treatment outcome is unclear.

Thus, it is currently difficult to establish 1) if machine learning can predict individual patient's symptom-related treatment outcomes with a clinically relevant accuracy, and 2) how to use machine learning regarding the many choices on variable selection, handling of missing data, and type of algorithm. Moreover, it is hard to gauge the consistency of machine learning algorithms across studies and contexts since they differ in time points for prediction, which conditions the patients are treated for, and model validation practices.

In the present study, we first aim is to evaluate the accuracy of machine learning outcome prediction, using a large sample of patients who have received ICBT in regular care, compared to a benchmark for clinical usefulness (67%)[19] and a simpler benchmark regression model using fewer variables. We find that the prediction models surpass the benchmark for clinical usefulness indicating clinical viability. Our secondary aim is to promote other researchers' future efforts in building predictive models, by comparing three primary aspects of methodological choices in machine learning: how to select predictive variables, how to handle missing data, and what machine learning algorithm to choose. In addition, we check consistency of models' performances over different time points for prediction, different conditions being treated, and different ways to validate algorithms. We find that no algorithm or combination showed clear superiority. These results indicate clinical usefulness for prediction models in predicting symptom outcomes in regular psychiatric care and that given the current sample size simpler models can be on par with more advanced.

## Methods

### Participants
The cohort were routine care patients from the Internet psychiatric clinic in Stockholm, who after psychiatric assessment[23–25] had received 12 weeks of ICBT for major depressive disorder, panic disorder or social anxiety disorder between January 2008 and November 2019. After patients with no outcome data were imputed, $n = 1017(15\%)$, 6695 patients were included. Of these, 3076 (46%) had received treatment for depression, 1767 (26%) for panic disorder, and 1852 (28%) for social anxiety disorder. This study received ethical approval from the Swedish ethical review authority Stockholm (Dnr: 2011/2091-31/3, amendment 2016/21-32, 2017/2320-32 and 2018/2550-32). This included the opt-out consent routine that is used for patients in the routine healthcare service including the internet psychiatric clinic in Stockholm. All patients at the Internet Psychiatry Clinic received information that their data could be used for research purposes and were given the opportunity to opt out.

### Treatments
ICBT at the clinic consists of 12 weeks of text-based self-help material, based on established CBT techniques for each condition, and has shown positive results[23–25]. It includes exercises, homework, asynchronous written conversations with therapists, and weekly self-assessments of primary symptoms. Therapists are licensed psychologists with CBT training and peer supervision. The clinic has been active since 2007 and the treatments and clinical guidelines for psychologists are largely the same[5].

### Treatment outcome
Primary symptoms were measured with Montgomery-Åsberg Depression Rating Scale-Self report (MADRS-S)[26] for depression, Panic Disorder Symptom Scale-Self Report (PDSS-SR)[27] for panic disorder, and Leibowitz Social Anxiety Scale-Self Report version (LSAS-SR)[28] for social anxiety disorder. All models predicted a standardised continuous outcome value for each patient. After the continuous outcome was predicted this score was dichotomized into 'success' if the score was below remittance for the scale or if sufficient symptom reduction indicated treatment response, or else as 'failure'. As such a dichotomisation of the outcome prediction was constructed after the continuous prediction had been made and a dichotomisation of the outcome itself was never conducted. The outcome value for each patient was assessed after the completion of the 12 weeks of treatment. This definition of the dichotomisation was used to prioritise finding patients at risk of failing treatment which is a core aspect of an Adaptive Treatment Strategy. We used previously established remitter scores which were 11 for MADRS-S[29], 8 for PDSS-SR[30] and 35 for LSAS-SR[31]. Sufficient symptom reduction was defined as 50% reduction from pre-treatment[32]. Dichotomisation was made for the exclusive reason to facilitate comparisons to other predictive models in the field and to reflect a possible clinical guidance in line with the 67% benchmark[19]—other continuous metrics were also calculated.

### Evaluation of predictive performance and clinical usefulness
The metric used for primary evaluation of a model's performance was balanced accuracy with 95% confidence intervals. Balanced accuracy was chosen to handle uneven distributions of success/failure and to simplify interpretation. $r^2$ was also used, as well as other metrics, see supplementary data 3. The benchmark for clinical usefulness, a balanced accuracy of 67%, was used since that was the balanced accuracy found in a handmade decision tree-type classification algorithm that was the key part of an adaptive treatment strategy shown in a randomized controlled trial to be successful in reducing the number of failed treatments in ICBT[19]. This indicates that this level of predictive accuracy was enough to enable clinical decisions and actions, which in the end was shown as beneficial for patients whose therapists used the adaptive treatment strategy compared to those that did not[2].

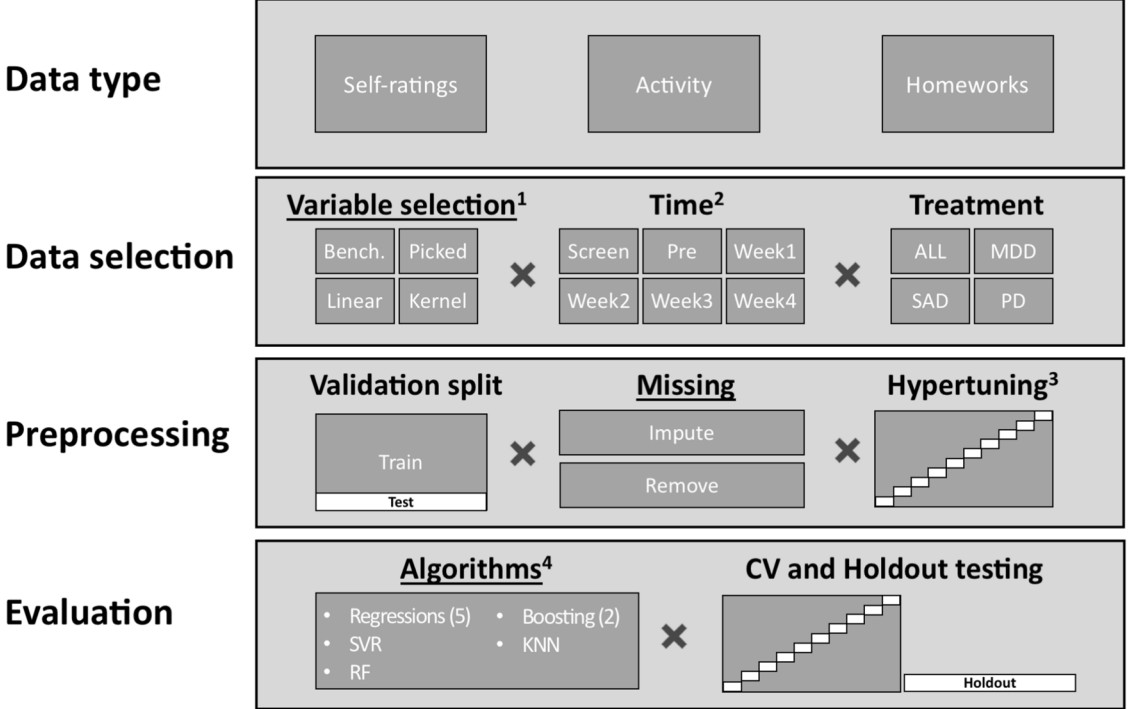

**Fig. 1 | Methodological pipeline and methodological choices when building prediction models.** Underlined headings designate primary aspects. Abbreviation: ALL, patients from all three treatments (only used together with handpicked and benchmark variable selection). MDD Major Depressive Disorder patients. SAD Social Anxiety Disorder patients. PD Panic Disorder patients. CV, 10-fold Cross-Validation. 1) Bench, Benchmark minimal dataset using only age, gender and primary symptom variables. Picked, Variables handpicked by clinical experts. Linear, Linear principal component analysis. Kernel, Kernel (radial basis function) principal component analysis. 2) Each time point also includes data from the previous points in time 3) Hypertuning performed for all algorithms using datasets from week four, with the exception of linear regression, and also of the kernel PCA number of components for the relevant datasets. 4) Regressions included: Ridge Regression, Elastic Net Regression, Lasso Regression, Bayesian Ridge Regression, and the benchmark method of Linear Regression, KNN K-Nearest Neighbour, RF Random Forest, SVR Linear Support Vector Regression. Boosting included: AdaBoost Regressor (utilizing random forest), Gradient Boosting Regressor.

## Prediction models and choices regarding model design

The design of a machine learning prediction model includes several methodological choices, visualised in Fig. 1. We examined how choices in relation to the three primary aspects of variable selection (four options), missing data management (two options), and algorithm selection (ten options) impacted performance. We also performed sensitivity analyses by investigating consistency of the models' performances across six-time points of prediction, three condition-specific treatments (or all patients combined), and two model validation procedures. In total, 1680 predictive models were built, analysed, and compared.

## Data type

Three types of data were used as predictors: self-rated assessments from the patients, logs of patient activity during treatment (e.g. the length and frequency of messages between therapist and patient) and extracted information from patients' homework reports. The homework variables were extensively feature-engineered through the construction of a representation of the text answers using the mean representation of a normalised vector created by Word2Vec[33], see supplementary method. All intervals and continuous variables were standardised (centred around 0 with a standard deviation of 1) to enable better model performance, merging of variables and data-driven variable selection. Nominal data were dummy transformed. All time variables, such as the day of the week a questionnaire was filled in, were transformed using a cyclic transformation, see equation 1. Equation 1. $Day = \sin\left(\frac{Day_{[0,6]}}{6}2\pi\right) + \cos\left(\frac{Day_{[0,6]}}{6}2\pi\right)$.

This was done to make Sundays more akin to Mondays. Such a transformation makes it easier for algorithms to find patterns in time variables. Treatment was dummy-coded in the dataset that includes all patients.

## Data selection

**Variable selection (first primary aspect).** Due to the large number of variables accessible we had to select which ones to include in the models to increase predictive performance. Variables were selected using four different approaches. 'Handpicked' variables were handpicked by experts in the field (authors) focusing on prediction in relation to outcome and limiting the number of features. The selection was done by: a) including variables that indicate activity in the treatment (e.g. what day of the week patient was active, number of messages, duration to fill in questionnaire), b) including variables important to symptom outcome (e.g. the sum of the primary symptom questionnaires) and other demographic variables, c) including variables deemed important to treatment and outcome (e.g. how well the homework's were completed). 'Benchmark' variables were a subset of the handpicked variables, consisting only of the weekly primary symptom measure, gender and age. Two Principal Component Analyses (PCA) were used as data-driven selection strategies. This was done to (unsupervised) keep as much information as possible from the entirety of the original data while making it useful for prediction. PCA was based on all information from the 12 most used questionnaires (including symptom measures) in addition to logs of activity and demographic information. After division of datasets by time and treatment, only variables with less than 25% missing were retained. 'Linear PCA' was conducted to retain 95% of the variance in the data, the resulting number of components was used as the predictors in the model and the number of components varied between treatments and datasets. 'Kernel PCA' used a radial basis function hyperparameter-tuned for optimal number of components (but not above 100) which were used as predictors for each algorithm and the selected number varied between treatments and datasets. For a complete overview of the variables and their distribution, see supplementary data 1.

**Time of prediction**. We used six-time points for prediction, and at each time point, all available variables for the designated variable selection were used to predict the treatment outcome which was based on the post-treatment score. The included times were: screening, pre-treatment, and treatment weeks 1-4. These were chosen since early prediction is key in Adaptive Treatment Strategies and later predictions might not provide enough time for adaptions[15]. Balancing the expected level of accuracy and time left to act before treatment ends, week four was chosen as the primary point of prediction.

**Treatment**. Each set of analyses was made separately for each condition-specific treatment, and the handpicked and benchmark variables were also used for another set of analyses for all patients aggregated. When aggregating patients into one dataset 'All' there was insufficient number of variables (e.g. different questionnaires) to enable the dataset to include the PCA selection strategies.

## Pre-processing
**Validation split**. The datasets were split into training (90%) and test (10%) datasets, with training of algorithms and imputation of data based on the training datasets. Furthermore, the training dataset implemented a 10-fold cross-validation. This was done to prevent overfitting and thus get more generalisable results.

**Handling missing data (second primary aspect)**. All predictor variables with more than 25% missing values after splitting the datasets by time, treatment, data selection and validation split were excluded. Before the level of missingness was explored exceptions were made for homework variables, comorbidities, and employment. These exceptions were made due to their expected high predictive value, and were only made in the imputation datasets, The exception was only used for the homework variables with 32–78% missing. The other two had 6% and 3% missing respectively and thus would not have been excluded even if the exception rule would not have been applied. During treatments worksheets are not mandatory to fill in, explaining the higher percentages of missing for the homework variables. Two strategies were then used to handle the remaining missing data. Firstly, missing data were imputed with Missforest[34] in Python using ten iterations to impute with all other variables as predictors in a Random Forest, which is a previously used strategy[35]. The imputation procedure also included and imputed the outcome in line with existing recommendations[36,37] having empirically shown that this gave better estimates and is not negatively affected by circularity, i.e. only confirming existing predictor-outcome relationships. Secondly, case removal was used, resulting in datasets with fewer patients. Finally, a sensitivity set of datasets was created for analyses where the outcome was not imputed, but predictors were.

**Hyperparameter tuning**. After splitting datasets as described above, hyperparameter tuning was based on datasets from week four for each algorithm ($n = 280$ datasets), with grid search and 10-fold cross-validation with $r^2$ as criterion for best fit. Due to the many datasets and the resulting combinatorial explosion nested cross-validation was not used. Hyperparameter tuning lets algorithms test out different options to see which results in the best fit for this dataset. For tuning parameters, see Supplementary Notes 1.

## Evaluation
**Algorithms (third primary aspect)**. Nine commonly used algorithms were trained and evaluated: lasso regression, gradient boosting regressor, elastic net regression, K-nearest neighbour, random forest, AdaBoost regressor (utilising random forest), linear support vector regression, ridge regression, Bayesian Ridge regression. Moreover, linear regression was used as a benchmark algorithm.

Python 2.7.1 and scikit-learn[38] were used for the implementation of the algorithms and their evaluation. See supplementary methods for more information on software used.

**Testing procedure**. To avoid overfitting, all metrics were tested both using 10-fold cross-validation (an additional cross-validation round separate from the hyperparameter cross-validation) as well as a separate testing dataset, withheld from the hyperparameter tuning, using a random sample of 10% from the initial sample. To facilitate generalisability and estimate variation of estimations the results from the 10-fold cross-validation are reported unless otherwise specified.

### Data, materials and code
The data for all results are in supplementary data 3. Specifications of predictors and datasets are specified in supplementary data 1. The analyses were not preregistered. Code for analysis of the prediction results and reproduction of the results in the paper can be found via link in the supplementary information and supplementary software 1.

### Reporting summary
Further information on research design is available in the Nature Portfolio Reporting Summary linked to this article.

## Results
### Participants
See Table 1 for baseline characteristics of all patients included in the analyses.

### Model development
The main results for the 1680 predictive models are presented below. Supplements contain distribution of each predictor and outcome for all datasets (Supplementary Data 1), hyperparameters investigated during tuning (Supplementary Notes 1), hyperparameter used (Supplementary Data 2), outcome in the datasets across both validation methods (supplementary data 3), additional results of tested aspects (supplementary results), code for the prediction procedure[39](Supplementary Software 1 and supplementary information for repository link) and code to reproduce the reported results (Supplementary Software 1 and supplementary information for repository link). To enable future comparisons, full results for all models including other metrics are reported in Supplementary Data 3, explanation of columns is found in Supplementary Notes 2.

### Overall predictive accuracy and benchmark comparisons
Table 2 shows the mean of the algorithms' performance (excluding linear regression) when applied on each treatment group datasets using the handpicked predictors up until week four, with imputation of missing data. The means are all clearly above the clinical benchmark of 67%, also for each smaller, condition-specific patient sample. The highest balanced accuracy score achieved was (mean [SD]) 80.7% [2.2%] with $r^2 = 0.599$ [0.0265] using random forest and handpicked data with the pooled patient sample. For detailed results showing all algorithms, all types of data selection, and the two types of missing data management, see Fig. 2.

The benchmark model, consisting of linear regression and the benchmark minimal set of predictors (primary symptom variables, age and gender), had a balanced accuracy of 77.8% [1.74%] and an $r^2$ of 0.538 [0.032] at week 4 for all patients (see Supplementary Model 1 in supplementary results for coefficients). The mean balanced accuracy across all algorithms, excluding linear regression, using benchmark data from week four, all patients, and imputation was 77.9% [1.64%] and the mean $r^2$ was 0.541 [0.0207]. Using handpicked variables instead of only benchmark data makes the accuracy increase for linear regression to 79.0% [1.82%] and an $r^2$ of 0.575 [0.0271]. Which is similar in performance to the random forest accuracy of 80.7% [2.2%].

**Table 1 | Baseline characteristics of patients included in analyses**

| Characteristics | Depression | | Panic | | Social | | Total | |
|---|---|---|---|---|---|---|---|---|
| | Female | Male | Female | Male | Female | Male | Female | Male |
| No. of patients (%) | 2027 (66) | 1049 (34) | 1077 (61) | 690 (39) | 1049 (57) | 803 (43) | 4153 (62) | 2542 (38) |
| Symptom level[a], mean (SD) | 23.31 (6) | 22 (7) | 11.61 (5) | 10.96 (5) | 72.94 (24) | 70.72 (24) | – | – |
| Age, mean (SD) | 37.18 (12) | 38.9 (13) | 33.47 (10) | 34.81 (11) | 32.31 (10) | 32.71 (10) | 34.99 (11) | 35.83 (12) |
| Married, No. (%) | 1089 (54) | 560 (53) | 713 (66) | 427 (62) | 590 (56) | 365 (45) | 2392 (58) | 1352 (53) |
| Employment status, No. (%) | | | | | | | | |
| Working | 1482 (75) | 761 (75) | 766 (74) | 551 (81) | 703 (69) | 558 (72) | 2951 (73) | 1870 (75) |
| Student | 213 (11) | 93 (9) | 164 (16) | 57 (8) | 202 (20) | 126 (16) | 579 (14) | 276 (11) |
| Other | 285 (14) | 166 (16) | 112 (11) | 71 (10) | 116 (11) | 96 (12) | 513 (13) | 333 (13) |
| Education level, No. (%) | | | | | | | | |
| Primary | 117 (6) | 72 (7) | 103 (10) | 74 (11) | 80 (8) | 74 (10) | 300 (7) | 220 (9) |
| Secondary | 859 (43) | 521 (51) | 542 (52) | 369 (54) | 457 (46) | 407 (53) | 1858 (46) | 1297 (53) |
| Postsecondary | 999 (51) | 425 (42) | 402 (38) | 238 (35) | 462 (46) | 283 (37) | 1863 (46) | 946 (38) |

[a]Symptom is the primary symptom measure for each treatment before treatment start (PRE). Montgomery-Åsberg Depression Rating Scale Self-report, Panic Disorder Symptom Scale-Self Report, and Leibowitz Social Anxiety Scale-Self report, respectively.

**Table 2 | Mean predictive models' performance across all algorithms (except linear regression) at week four in treatment**

| | Depression | Panic | Social | All |
|---|---|---|---|---|
| No. of patients | 2768 | 1590 | 1667 | 6026 |
| Balanced accuracy, mean % (SD) | 74.42 (2.74) | 73.69 (2.56) | 74.03 (3.89) | 78.17 (2.26) |
| $R^2$, mean (SD) | 0.50 (0.05) | 0.49 (0.04) | 0.67 (0.05) | 0.56 (0.04) |

All models used data from handpicked variables and imputation of missing data.
SD standard deviation.

### Predictive accuracy dependent on the primary aspects of variable selection, missing management, and algorithms

Figure 2 presents an overview of the balanced accuracies for different choices of the three primary methodological aspects, i.e. selection of variables, missing data management, and type of machine learning algorithm, applied to the four different patient samples.

### Variable selection

In general, the expert handpicked-variable datasets outperformed the other selection methods with scores of benchmark and handpicked datasets at week 4 being (mean [SD]) 71.9% [5.09%] and 71.8% [5.76%] accurate respectively compared to the linear and kernel PCA with means 67.1% [5.91%] and 61.1% [8.94%], see Fig. 3. However, at pre-treatment these scores were more homogenous: 62.7% [5.53%], 62.8% [5.43%], 61% [5.61%], 57% [6.89%] for benchmark, handpicked, linear PCA, and kernel PCA datasets respectively, showing that the discrepancy in results between handpicked methods and PCA methods increased over time. Albeit the variance of accuracy for the different variable selections was most evident when combined with certain algorithms. This interaction was prominent with the kernel PCA selection of variables, which performed worse than the other selection methods, especially for lasso regression and elastic net regression: without these, the kernel PCA method had a higher, but still the lowest, mean at week 4: 64.6% [6.17%].

### Handling missing data

Across all datasets, imputing data resulted in a higher balanced accuracy. The mean difference in balanced accuracy (mean 95% CI [LB, UB]) 4.29% [3.59%, 4.98%] between using case removal (60.9% [7.00%]) and imputation (65.2% [7.52%]). Data loss was considerable using case removal (mean [min-max]) 54.89% [28.25–85.9%], and the biggest

decrease in accuracy was at week 4 using case removal (65.8% [7.45%]) compared to imputation (71.3% [7.00%]). The sensitivity analysis where the missing outcome data were not imputed but removed revealed a slightly lower overall balanced accuracy, 62.2% [6.74%] for predictor-only imputed dataset. The best-performing model was still random forest using handpicked features and all patients at week four, but with a 76% [1.61%] balanced accuracy, and the score for linear regression for the same dataset was 75.5% [1.88%]. Finally, the mean balanced accuracy for predictor-only imputed datasets using the handpicked predictors up until week four was: 69.41% [2.31%] for depression, 68.06% [3.04%] for panic disorder, 70.20% [3.17%] for social anxiety disorder, and 74.62% [1.64%] for 'All' patients aggregated.

### Algorithm

Overall, there were some variances across models. There was a disadvantage for models with variable weighting, such as lasso regression and elastic net regression, when used together with kernel or linear PCA selection of predictors, see Fig. 4. As can be seen in Fig. 4 there were no clearly superior algorithms even accounting for this interaction between models with variable weighting and PCA selection. Excluding the PCA variable selection strategies increased the mean [SD] lasso regression accuracy from 65.2% [11.1%] to 71.9% [1.33] and elastic net regression from 66.6% [11.7%] to 73.6% [1.14%]. This exclusion also made random forest the best-performing model, increasing its mean [SD] balanced accuracy from 72.1% [5.09%] to 76% [0.94%]. This resulted in a small significant mean difference (mean 95% CI [LB, UB]) of 1.81% [0.53%, 3.09%] between random forest (76% [0.94%]) and the linear regression benchmark (74.7% [1.04%]). However, without the exclusion of the PCA selection, this mean difference was non-significant (mean 95% CI [LB, UB]): −0.27% [−4.07%, 3.54%].

### Sensitivity analyses

The sensitivity analyses showed consistency of the three primary aspects across time points, treatments, and validation methods. The predictive performance increased over time, exemplified with the top performing random forest model starting with 71.8% [1.66%] balanced accuracy at screening, increasing to 72.6% [1.43%] at pre-treatment, and to 80.7% [2.21%] at week four in the pooled patient sample. See supplementary results and supplementary fig. 1 for more details. Table 2 shows that the algorithms were similarly accurate across the different treatments but pooling all conditions increased accuracy and decreased variance. Finally, there was a negligible difference on balanced accuracy 1.44% [4.72%] between cross-validation (63.1% [7.57%]) and holdout testing (64.5% [8.85%]) across all 1680 models.

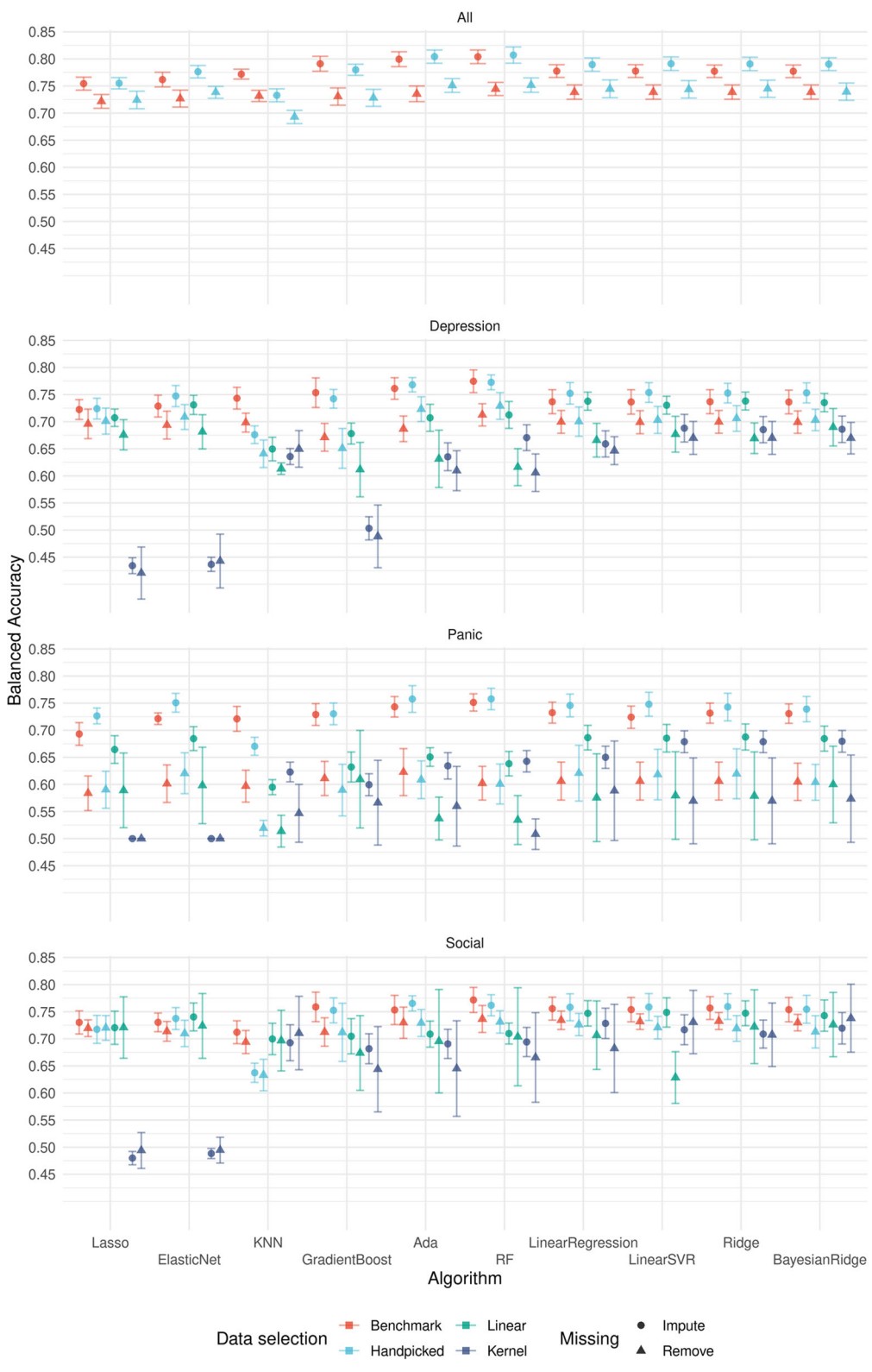

**Fig. 2 | Balanced accuracy for different methodological choices.** Balanced accuracy (CI-95%) for the 280 prediction models exploring all combinations of the three primary methodological aspects (Data Selection, Handling of Missing Data, Machine Learning Algorithm) for all four patient samples at the primary point of prediction at treatment week four, validated with 10-fold cross-validation. Varying machine learning algorithms, variable selection methods, and how missing data was handled. Divided into different treatments in addition to the aggregated data (All). Error bars indicate 95% confidence intervals based on $n = 10$ cross-validation. For Lasso and Elastic Net for Panic Disorder with kernel data selection, there was no variation of results across the validation sets. For the 'All' dataset the PCA selection strategy was not available due to the limited number of aggregated variables across different treatments. Abbreviation: Lasso Lasso Regression. GradientBoost Gradient Boosting Regressor. ElasticNet Elastic net regression. KNN K-Nearest Neighbour. RF Random forest. Ada AdaBoost Regressor. LinearSVR Linear Support Vector Regression. Ridge Ridge Regression.

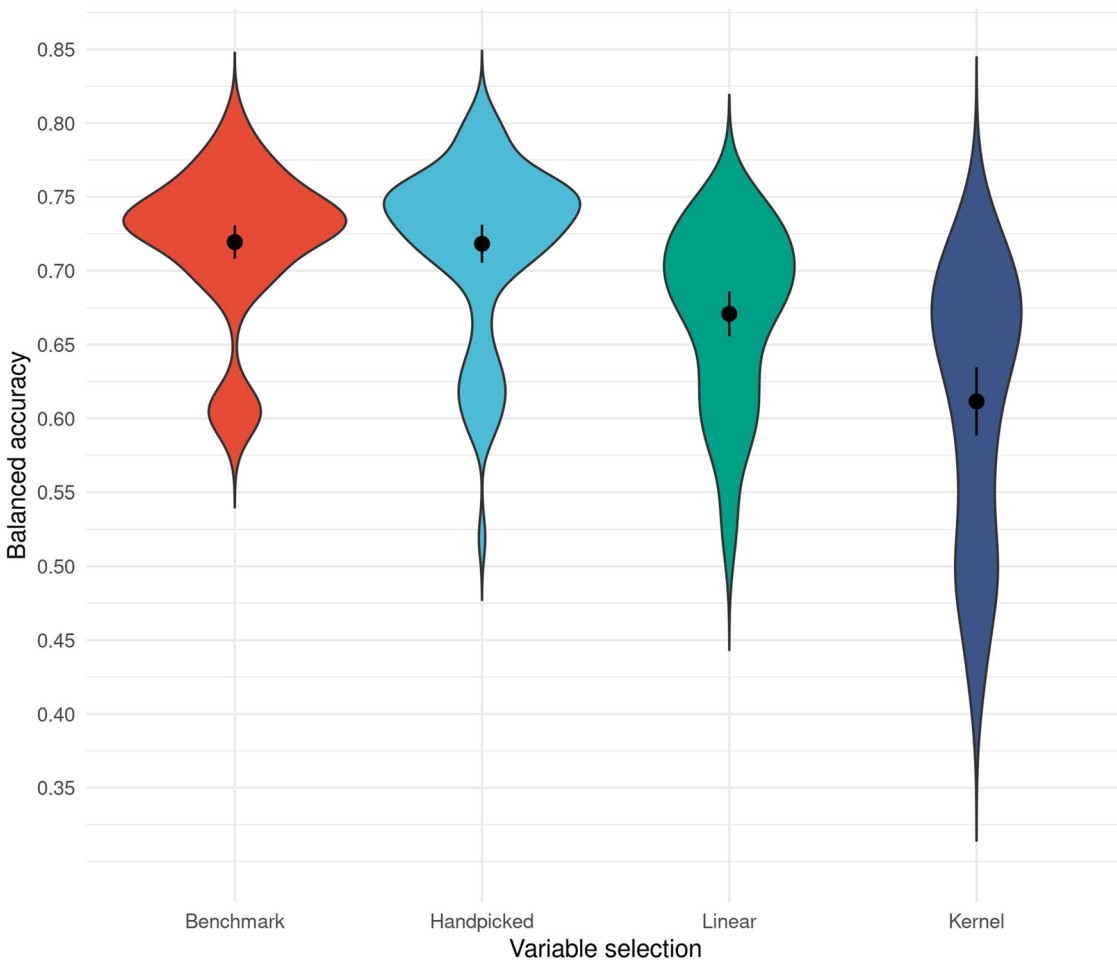

**Fig. 3 | Mean balanced accuracy for variable selection.** Overall mean balanced accuracy (CI-95%) and spread for the different variable selections on a total of 280 different dataset configurations at treatment week four. Data divided by variable selections while looking at prediction at week 4 in treatment while varying machine learning algorithm ($n = 10$), and how missing data was handled ($n = 2$) as well as what treatment data was used ($n = 4$ for benchmark/handpicked, $n = 3$ for Linear/kernel)). Mean dot and error bars indicate mean and 95% confidence intervals. Abbreviation: Linear Linear principal component analysis. Kernel Kernel principal component analysis.

### Synthesis: methodological choices for the best predictive model

The best model was a result from the interaction between the three primary aspects at week four. While the benchmark set of predictors and the expert handpicked variable set had similar means for their overall accuracies, the highest balanced accuracy was attained using handpicked variables with random forest. Overall, the datasets with imputed data had a higher accuracy score and this was also used in the model with the highest attained balanced accuracy. The interaction between these three choices for the primary methodological aspects is shown in Fig. 5 as the resulting best predictive model applied on all four samples at week four. While this combination performed best, it cannot be deemed decidedly superior as there were other close contenders.

### Discussion

Our first aim was to evaluate the clinical usefulness of machine learning predictions. We found that the mean balanced accuracy across machine learning algorithms was 78.1% when imputing outcomes and using handpicked variables from the dataset including all patients at week four in treatment. The corresponding figures for each specific treatment varied between 73.7% and 74.4%; all above the 67% benchmark for clinical usefulness[19]. This clearly indicates that machine learning predictions could be clinically useful if used to detect patients at risk of failure within an Adaptive Treatment Strategy. Interestingly, the top performing model using random forest surpassed the 67% benchmark already at week 2 in each

treatment showing a potential for predictions earlier than week four in treatment. However, no machine learning algorithm was clearly superior to traditional statistics, i.e. the linear regression benchmark, and lasso regression did not surpass the benchmark. The top balanced accuracy was 80.7% using random forest in the pooled sample, compared to 79% for linear regression and 77.8% using the simpler benchmark model with fewer predictors.

Our second aim of evaluating different methodological choices to promote future efforts was based on an unprecedented empirical evaluation of a large number of machine learning models, varying a range of methodological aspects, on a large, high-quality clinical sample. There was some variation across the three primary aspects we explored. The first aspect, on how to select predictor variables, showed that the set of variables handpicked by experts outperformed the two data-driven selections (PCA). The PCA selections also had a clear negative interaction with certain algorithms. Previous research has shown that PCA can lower the accuracy of classification with medical data[40]. PCA creates independent variables, which affects the correlation between the symptom variables. Symptom variables correlate strongly with the outcome, and creating independent variables results in a worse performance. Our results support this, since datasets from treatment start, which contain fewer correlated variables, do not show this discrepancy in results between selection methods. Many feature selection methods in the field were not tested, such as recursive feature elimination, LASSO or Elastic-Net. Additionally, other research indicates a use for probabilistic

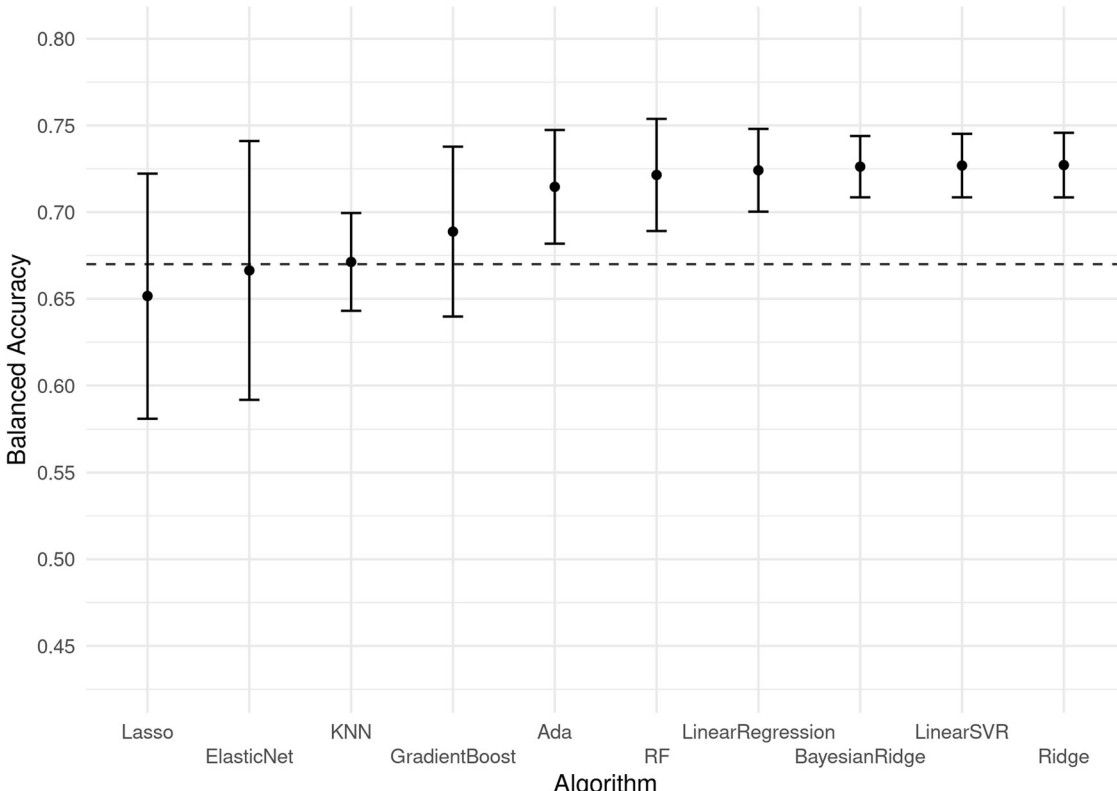

**Fig. 4 | Mean balanced accuracy for algorithms.** Mean balanced accuracy (CI-95%) for different algorithms on 120 different dataset configurations at treatment week four using imputed data. Varying variable selection, method and treatment dataset —excluding the 'All' dataset. Error bars indicate 95% confidence intervals ($n = 12$). Dotted line indicates the benchmark for clinical usefulness: an accuracy of 67%.

Abbreviation: Lasso Lasso Regression. GradientBoost Gradient Boosting Regressor. ElasticNet Elastic net regression. KNN K-Nearest Neighbour. RF Random forest. Ada AdaBoost Regressor. LinearSVR Linear Support Vector Regression. Ridge Ridge Regression.

latent models[41] to classify patients based on patterns over time and then relate the classes to outcome.

The second aspect, on how to best handle missing data, showed a clear difference with case removal having a lower accuracy while also removing considerable amounts of data. When imputing only predictor data the mean balanced accuracy across machine learning algorithms was lower 74.6% instead of 78.1%, this was also reflected in the top-performing combination using random forest which accuracy was 4.7% lower and the linear regression which was 3.5% lower. Previous research shows that imputation (including outcome) provides more robust estimates than case removal, that differences in estimates between methods are not necessarily interpretable[37] and does not create circularity in the data[36]. Because of this, the mean difference in accuracies is hard to interpret, but the conservative estimate (of not imputing outcome) still gives balanced accuracies above the clinical benchmark of 67% for all treatment groups.

Regarding the third aspect, no overall superior machine learning algorithm was found, but the lasso regression and elastic net regression had a lower accuracy in several datasets due to the interaction with variable selection strategies.

Finally, our sensitivity analyses showed no distinct inconsistency for the main aspects besides the interaction between time and PCA variable selection mentioned above. Overall, the analyses showed an expected linear increase in accuracy later in treatment, in line with previous research the baseline models (before treatment started) did generally not achieve the predefined clinical usefulness threshold of 67% especially not in the treatment-specific subsets. There was a greater accuracy when pooling all patients, and a negligible difference between 10-fold cross-validation and holdout testing. This latter difference seems to indicate that the models did not overfit during hyperparameter tuning.

One purpose of our extensive comparison between large numbers of models was to help guide future machine-learning efforts with aims similar to ours. The resulting guidance would be to let clinical and statistical experts carefully consider which predictive variables to include, to impute missing data, and evaluate a benchmark model (e.g. linear regression) together with a robust machine learning algorithm such as random forest. These are not definite recommendations but rather findings to guide future research since we have only applied our models to one clinical context and found limited variance in our results. While numerous analyses were conducted, other choices in each of the main aspects could have been made. Still, we believe that these findings can help in building future machine learning applications for mental health, especially for similar contexts where guided ICBT is used, as this type of treatment is increasingly disseminated in clinics serving an increased number of patients worldwide[5]. Future research could also investigate transfer learning between such clinics to test generalisability. An important aspect of our findings is that it might also support future decisions to not spend resources on building machine learning models but instead opt for simpler models based on traditional statistics. Additionally, since the imputation method (more discussed under Limitations) used random forest, this method could have been favoured compared to other algorithms. This further highlights the need for careful consideration of the context, quality, and volume of the data for prediction research in psychiatry. Further research should focus on what conditions machine learning algorithms need to thrive in psychiatric settings and psychological treatments[42].

This study includes an empirically based benchmark for when accuracies reach a good enough level to be clinically useful within an Adaptive Treatment Strategy for psychological treatment[2,19]. We succeeded in reaching above this benchmark, but our comparison of different models indicates that machine learning algorithms with more variables were not

**Fig. 5 | Balanced accuracy for best methodological choice.** Balanced accuracy for the best combination of methodological choices (Random forest, hand-picked variables and imputed missing data) at week 4. Datasets used handpicked variables, imputed missing data, used random forest as algorithm. Dotted line indicates the clinical benchmark accuracy of 67%. Abbreviation: CV cross-validation (blue square), HO hold-out (orange diamond). Error bars indicate 95% confidence intervals based on $n = 10$ cross-validation.

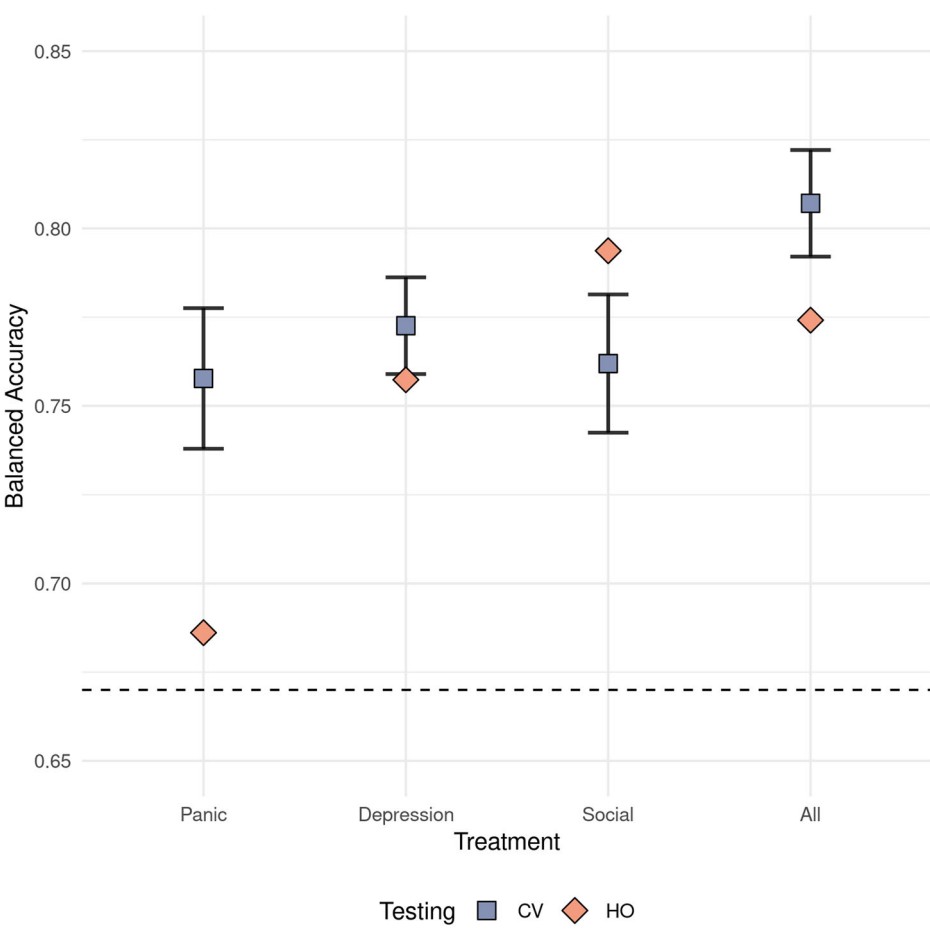

necessarily superior to a simpler linear model with a few variables. The lack of superiority for machine learning methods was not limited to the treatment-specific datasets but also in the larger dataset combining treatments. This finding is supported by Christodoulou and colleagues, who showed no difference between machine learning models for different clinical conditions and a benchmark (logistic regression)[8]. Thus, implementing machine learning methods in clinical psychology does not necessarily lead to superior predictions[43] even in datasets with around 6500 cases. This finding constitutes an important reminder that omitting benchmark comparisons can lead to false optimism[20]. This also aligns with studies on traditional psychological treatment showing that simple statistical, non-machine learning, predictive models applied on repeated symptom measures during treatment can provide clinically useful guidance[1,3,10,15,44]. Studies using larger sample sizes with sound methodology to predict outcomes for patients, both from regular care[45] and ICBT[35], show accuracies on par with our results, while smaller sample sizes have produced biased accuracies[21]. Hence, it is expected that our predictive models show a lower accuracy compared to some studies using machine learning in clinical samples[9], but the results from our analyses are more robust. These results also highlight that traditional statistics can also be a clinically useful prediction model, requiring fewer variables and resources, even though machine learning still holds the potential to increase predictive accuracy.

There are some limitations to this study. We imputed missing data, including the outcome, and while we consider this a valid approach according to recommendations[36,37] it could have influenced our scores. The sensitivity analysis indicates that imputing predictors but not outcomes decreased accuracy by 3.5% on average. We cannot know if this increase in prediction accuracy for outcome imputation was due to the procedure introducing bias despite the previous findings to the contrary[36,37], or if it was because the impute-all approach was superior in a non-biased way, and it could be a combination of both. A multiple imputation procedure could

have further ensured robustness of our results and could be a way to quantify this uncertainty but was deemed too computationally costly due to the combinatorial increase of analyses multiple datasets would entail. However, both Moons et al. [36] and van Ginkel et al., [37] underscores the preference for including outcome in imputation procedures. Also, while we report the continuous performance measures (e.g. r2) the binary outcomes are influenced by the chosen remitter and responder criteria as has been shown in ref. [2]. Another limitation is that even though an empirically established based accuracy of 67%[19] for clinical usefulness was used, it is not necessarily generalizable over patient conditions, interventions, and clinical contexts. In that study, the clinical context, i.e. ICBT and the way it was delivered, was very similar, but the patients were treated for Insomnia. We assume an improved accuracy is always more beneficial in an adaptive treatment strategy, but it is more difficult to establish a decisive cut-off for when it is good enough and this needs to be further explored. However, our chosen benchmark is also quite similar to the cut-off of 65% previously reported by clinicians to be perceived as acceptable to act upon for clinical decisions in general[18]. We have not explored which specific variables were most useful for prediction since it was outside the scope of our aim, but our results indicate that weekly symptoms have a large impact on results, see Supplementary Data 1 for included variables. We did not use nested cross-validation for hyperparameter tuning which increase the risk of overfitting, this was due to computational constraints. However, our results showed a negligible discrepancy between cross-validation and holdout scores, indicating small overfitting if any. We used a grid search for hyperparameter tuning. While we think this was sufficient, we have not exhausted the possibility to find better parameters which might have increased the performance of some of our machine-learning models.

The results indicate a good enough predictive performance of evaluated machine learning models to be used within Adaptive Treatment Strategies applied at treatment week four in ICBT. The recommended predictive

model with handpicked predictors imputed missing data, and using random forest indicated clinically useful predictive accuracy also at week two.

Contrary to several previous studies, when we applied proper validation and rather large training datasets to minimize inflation of accuracy, there was not any clear superiority for complex prediction algorithms or data selection methods. In a machine learning context, the dataset is still relatively small and as such future efforts with even larger training datasets and other methodological approaches might still favour machine learning. Although, it is encouraging that the simpler benchmark prediction model achieved an accuracy above the benchmark for clinical usefulness since simplicity facilitates implementation. While randomised clinical trials to directly evaluate the effects of Adaptive Treatment Strategies based on machine learning predictive models are warranted, our findings are highly encouraging regarding their clinical usefulness and also highlight the need to consider patient characteristics, the volume of data, and the selection of variables and algorithms, as well as to question if a more complex method always is better.

## Data availability

The data for all results are in Supplementary Data 3, this data also includes the source data to reproduce the figures in this paper. Specifications of predictors and datasets are specified in Supplementary Data 1. Underlying patient data is not in the supplement due to scope of the approved ethical application and current health care data management policy, and are available according to the ethical approval upon reasonable request.

## Code availability

Code for analysis of the prediction results and reproduction of the results in the paper can be found at: https://github.com/intraverbal/paper_ipsy_outcome_pred[39], also detailed in the supplementary information, also in the supplementary software 1. Code for analyses, imputation, and data set creation is also available in the supplementary information and supplementary software 1. Code for underlying handling of patient data is limited due to the scope of the approved ethical application and current health care data management policy.

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

## Acknowledgements

This work was mainly supported by The Swedish Research Council (VR), The Erling Persson family foundation (EP-Stiftelsen), and The Swedish ALF agreement between the Swedish government and the county councils, with additional funding by the Swedish Foundation for Strategic Research (SSF), Psykiatrifonden, and Thuring's Foundation. The funding sources were not involved in any part of the study.

## Author contributions

All authors contributed extensively to the work presented in this paper. N.H.I., F.B.A., E.F., M.B., and V.K. were involved in conceptualization and design, with N.H.I and V.K. leading. N.H.I. leads data management with support from F.B.A., E.F., M.B., V.K. N.H.I. lead the methodological development and F.B.A., E.F., M.B., V.K. supported in the research investigation. N.H.I. conducted all analyses. N.H.I., F.B.A., E.F., M.B., and V.K. contributed to writing and reviewing the paper with N.H.I leading.

## Funding

## Competing interests

The authors declare no competing interests.
