## [Peer Review File · Communications Medicine]

This manuscript has been previously reviewed at another Nature Portfolio journal. This document only contains reviewer comments and rebuttal letters for versions considered at *Communications Medicine*.Reviewers' comments:

Reviewer #2 (Remarks to the Author):

Many thanks to the authors for making the effort to submit a second revision of their work. Most of my criticisms have now been eliminated. The use of regression for prediction and the subsequent binarization procedure are now very clear and understandable.

I am still unconvinced by the multiple imputation of outcomes, but I agree that an advantage of the study is that the authors present results both with and without imputation. Maybe the authors can add a very short discussion of the overall issue in the discussion. This could largely be similar to their argument for the method as in this and the previous reviewer response, but with a short mentioning of the potential risk of introducing label noise. The reader can then decide for themselves which approach is more convincing.

Finally, it is great that the authors now decided to make their ML pipeline available. However, the code is really hard to read from a pdf. Please consider uploading to Github or a similar platform instead.

Reviewer #4 (Remarks to the Author):

This is a revised manuscript, for which I was not a reviewer before. The aim of the study was to compare machine learning (ML) models for outcome prediction in Internet-delivered psychotherapy. In naturalistic data of more than 6,000 patients, 80 models were trained and tested. Besides different ML algorithms, several variable selection methods were compared to each other, and missing value imputation (yes, no) was varied. To improve models, hyperparameter tuning and cross-validation were applied. An algorithm based on handpicked variables with missing data imputation was found to be superior to other approaches. However, the different ML algorithms did not vary meaningfully in their accuracy. The authors conclude that ML proved useful in predicting treatment outcome of Internet-based therapy, however, advantages of individual algorithms must be investigated in further research.

The study addresses an important and topical issue by examining the outcome of psychotherapy patients in regular care treated with Internet-based interventions. To date, there is only little research and knowledge on which ML algorithms are useful in which situation and with which data. Therefore, this study provides a guide for data analysis and on adequate algorithms and methodological choices. The study seems methodologically sound, and it is well-written. However, there are some major concerns that need to be addressed.

Major:

-pp. 0–2: The introduction seems somewhat superficial and does not consider the relevant literature on predictive models in psychotherapy research, especially for binary outcome variables. There are comparable recent papers, particularly in dropout prediction, that could be considered, e.g. Bennemann et al. (2022) and Giesemann et al. (2023). Furthermore, the M4 Competition presents several publications that are interesting in the context of such model comparisons (e.g., Makridakis et al., 2018a, 2018b). The suggestions do not necessarily have to be adopted, but overall, the introduction seems too incomplete and does not represent the current state of research to adequately introduce this topic.

Bennemann, B., Schwartz, B., Giesemann, J., & Lutz, W. (2022). Predicting patients who will drop out of out-patient psychotherapy using machine learning algorithms. *The British Journal of Psychiatry*, 220(4), 192–201. <https://doi.org/10.1192/bjp.2022.17>

Giesemann, J., Delgado, J., Schwartz, B., Bennemann, B., & Lutz, W. (2023). Predicting dropout from psychological treatment using different machine learning algorithms, resampling methods, and sample sizes. *Psychotherapy Research*, 33(6), 683–695.

<https://doi.org/10.1080/10503307.2022.2161432>

Makridakis, S., Spiliotis, E., & Assimakopoulos, V. (2018a). The M4 Competition: Results, findings, conclusion and way forward. *International Journal of Forecasting*, 34(4), 802–808. doi: 10.1016/j.ijforecast.2018.06.001

Makridakis, S., Spiliotis, E., & Assimakopoulos, V. (2018b). Statistical and machine learning forecasting methods: Concerns and ways forward. *PloS One*, 13(3), e0194889. doi: 10.1371/journal.pone.0194889

-p. 2, l. 89: As a benchmark for clinical usefulness, 67% accuracy was defined. Although a reference for this number is given, it is unclear whether 67% is really a good accuracy and has clinical utility. It should be described in more detail, why a 33% error rate should be good enough for clinical practice. Additionally, the base rate of the binary variable (50% symptom reduction yes vs. no) should be reported to evaluate the quality of the prediction algorithm and to see if the algorithm is better than predicting 0 or 1 for all patients.

-p. 18, l. 441: A 10% test sample seems quite small. When no k-fold- cross validation is applied, but data are split into train and test samples, a 70/30 split would be more common. I agree that large data are needed for training a good model, however, the reliability of the test depends on the sample size of the test data. Please elaborate on this point and explain the choice for a 90/10 split.

-p. 19, l. 447: Variables with more than 25% missingness were excluded. However, this rule was not applied to some variables, including homework, comorbidities, and employment. This inconsistent approach should be explained. Furthermore, the missingness of these variables should be reported in total cases and percentages. First, this is especially important for the approach, in which these missings are statistically imputed. Second, many missings in these variables would result in a small sample size for the approach without imputation. Therefore, the sample size for each analysis should be reported to provide readers with enough information to interpret the results appropriately (maybe imputing is just better because of the larger sample size).

-p. 19., l. 452: Although again there is a reference given, I'm quite sceptical about imputing the outcome variable together with its predictors. It seems obvious to me that this approach creates associations between predictors and outcome (especially in the case of a high rate of missingness; see my previous concern).

Minor:

-p. 2, l. 89: The second benchmark was a regression model with "fewer variables". Please specify what "fewer" means. How many variables were in the model and how were they selected?

I think that the structure (methods after discussion) is not beneficial to understand the manuscript. The results section in its current form is not clear without having read the methods. Therefore, I would recommend restructuring the manuscript. I know the structure will be given by the journal guidelines, however, in this case, maybe the authors or the editor should think about changing the structure.

Revision #3: Response to referees

Reviewer #2 (Remarks to the Author):

Many thanks to the authors for making the effort to submit a second revision of their work. Most of my criticisms have now been eliminated. The use of regression for prediction and the subsequent binarization procedure are now very clear and understandable.

I am still unconvinced by the multiple imputation of outcomes, but I agree that an advantage of the study is that the authors present results both with and without imputation. Maybe the authors can add a very short discussion of the overall issue in the discussion. This could largely be similar to their argument for the method as in this and the previous reviewer response, but with a short mentioning of the potential risk of introducing label noise. The reader can then decide for themselves which approach is more convincing.

Finally, it is great that the authors now decided to make their ML pipeline available.

However, the code is really hard to read from a pdf. Please consider uploading to Github or a similar platform instead.

Authors:

We thank the reviewer for another round of review and are happy to hear that the changes to the manuscript have improved the quality of our work. In regards to the imputation of outcome we agree that a discussion to reflect the possible issues is needed. As such, in our discussion and limitations we now updated to the following (marked in yellow):

“An important aspect of our findings is that it might also support future decisions to *not* spend resources on building machine learning models but instead opt for simpler models based on traditional statistics. **Additionally, since the imputation method (more discussed under Limitations) used random forest, this method could have been favored compared to**

other algorithms. This further highlights the need for careful consideration of the context, quality, and volume of the data for prediction research in psychiatry”

And in the limitations we further discuss the imputation:

“There are some limitations to this study. We imputed missing data, including the outcome, and while we consider this a valid approach according to recommendations (Moons et al., 2006; van Ginkel et al., 2020) it could have influenced our scores. The sensitivity analysis indicates that imputing predictors but not outcome decreased accuracy by 3.5% on average.

We cannot know if this increase in prediction accuracy for outcome imputation was due to the procedure introducing bias despite the previous findings to the contrary (Moons et al., 2006; van Ginkel et al., 2020), or if it was because the impute-all approach were superior in a non-biased way, and it could be a combination of both. A multiple imputation procedure could have further ensured robustness of our results and could be a way to quantify this uncertainty but was deemed too computationally costly due to the combinatorial increase of analyses multiple datasets would entail. However, both Moons et al., (2006) and van Ginkel et al., (2020) underscore the preference for including outcome in imputation procedures.”

It is unfortunate to hear that the reviewers received the code in the pdf format. We had also attached the code as a zip file. But, if it is possible for the journal we could upload the code in a suitable text format (.py) to the journal – as that should be the most suitable way to have the code attached to the paper. Otherwise, or perhaps as the best solution, we have also added the code on a github repository available at this link:

https://github.com/intraverbal/paper_ipsy_outcome_pred

This is also reflected by the following changes to the paper:

" Supplements contain distribution of each predictor and outcome for all datasets (supplement 1), hyperparameters investigated during tuning (appendix in supplement 2), hyperparameter used (supplement 3), outcome in the datasets across both validation methods (supplement 4), additional results of tested aspects (results in supplement 2), code

for the prediction procedure (supplement 2) and code to reproduce the reported results (supplement 5). "

and in the supplement 2:

" Code for prediction procedure is found on github on the following link:

https://github.com/intraverbal/paper_ipsy_outcome_pred"

Reviewer #4 (Remarks to the Author):

This is a revised manuscript, for which I was not a reviewer before. The aim of the study was to compare machine learning (ML) models for outcome prediction in Internet-delivered psychotherapy. In naturalistic data of more than 6,000 patients, 80 models were trained and tested. Besides different ML algorithms, several variable selection methods were compared to each other, and missing value imputation (yes, no) was varied. To improve models, hyperparameter tuning and cross-validation were applied. An algorithm based on handpicked variables with missing data imputation was found to be superior to other approaches. However, the different ML algorithms did not vary meaningfully in their accuracy. The authors conclude that ML proved useful in predicting treatment outcome of Internet-based therapy, however, advantages of individual algorithms must be investigated in further research.

The study addresses an important and topical issue by examining the outcome of psychotherapy patients in regular care treated with Internet-based interventions. To date, there is only little research and knowledge on which ML algorithms are useful in which situation and with which data. Therefore, this study provides a guide for data analysis and on adequate algorithms and methodological choices. The study seems methodologically sound, and it is well-written. However, there are some major concerns that need to be addressed.

Authors:

We extend our thanks to the reviewer for their time and effort spent on evaluating our paper.

Reviewer #4

-pp. 0–2: The introduction seems somewhat superficial and does not consider the relevant literature on predictive models in psychotherapy research, especially for binary outcome

variables. There are comparable recent papers, particularly in dropout prediction, that could be considered, e.g. Bennemann et al. (2022) and Gieseemann et al. (2023). Furthermore, the M4 Competition presents several publications that are interesting in the context of such model comparisons (e.g., Makridakis et al., 2018a, 2018b). The suggestions do not necessarily have to be adopted, but overall, the introduction seems too incomplete and does not represent the current state of research to adequately introduce this topic.

Bennemann, B., Schwartz, B., Gieseemann, J., & Lutz, W. (2022). Predicting patients who will drop out of out-patient psychotherapy using machine learning algorithms. *The British Journal of Psychiatry*, 220(4), 192–201. <https://doi.org/10.1192/bjp.2022.17>

Gieseemann, J., Delgadillo, J., Schwartz, B., Bennemann, B., & Lutz, W. (2023). Predicting dropout from psychological treatment using different machine learning algorithms, resampling methods, and sample sizes. *Psychotherapy Research*, 33(6), 683–695. <https://doi.org/10.1080/10503307.2022.2161432>

Makridakis, S., Spiliotis, E., & Assimakopoulos, V. (2018a). The M4 Competition: Results, findings, conclusion and way forward. *International Journal of Forecasting*, 34(4), 802–808. doi: 10.1016/j.ijforecast.2018.06.001

Makridakis, S., Spiliotis, E., & Assimakopoulos, V. (2018b). Statistical and machine learning forecasting methods: Concerns and ways forward. *PloS One*, 13(3), e0194889. doi: 10.1371/journal.pone.0194889

Authors:

We understand the reviewer's concerns and agree that the introduction could be somewhat expanded. There are several other outcomes in psychotherapy research which could be investigated, predicted and reviewed. Dropout prediction is not the topic investigated in the current paper, but that field is closely related and we have thus expanded the introduction with some of the suggested references. Changes are highlighted:

“Bennemann et al. (2022) identified that algorithms incorporating predictor selection and selecting from several variables, such as tree-based and boosted machine learning algorithms was beneficial for accurately predicting therapy dropout, an outcome distinct from symptomatic outcome yet likely linked. “

And:

“In general time-series forecasting studies, statistical and machine learning methods have shown different levels of accuracy (Makridakis, Spiliotis, & Assimakopoulos, 2018), with a higher accuracy for statistical methods: however, the results' applicability to prediction of psychological treatment outcome is unclear.”

We have now also, to clarify, further specified that it is symptom-related treatment outcome we are investigating in the following paragraphs in the introduction (changes highlighted):

“However, it is well established that early symptom change during treatment is associated with the symptomatic treatment outcome (Beard & Delgadillo, 2019; Szegedi et al., 2009) and including predictors from the first weeks of treatment, as in Routine Outcome Monitoring (Barkham et al., 2023) and Adaptive Treatment Strategies, increases the accuracy of predictions compared to baseline predictors only (Forsell et al., 2020; Hoogendoorn et al., 2017)”

“Thus, it is currently difficult to establish 1) if machine learning can predict individual patient's symptom-related treatment outcome with a clinically relevant accuracy,”

We have also previously cited papers from both authors designated by reviewer (Lutz and Delgadillo) in our introduction in relation to the symptom treatment outcome, and we now added one more to more clearly refer to the broad field of Routine Outcome Monitoring:

Barkham, M., De Jong, K., Delgadillo, J., & Lutz, W. (2023). Routine Outcome Monitoring (ROM) and feedback: research review and recommendations. *Psychotherapy Research*, 1-15.

Reviewer #4

-p. 2, l. 89: As a benchmark for clinical usefulness, 67% accuracy was defined. Although a reference for this number is given, it is unclear whether 67% is really a good accuracy and has clinical utility. It should be described in more detail, why a 33% error rate should be good enough for clinical practice. Additionally, the base rate of the binary variable (50% symptom reduction yes vs. no) should be reported to evaluate the quality of the prediction algorithm and to see if the algorithm is better than predicting 0 or 1 for all patients.

Authors:

We thank the reviewer for pointing this out, since we agree that our description of the clinical benchmark of 67% is a bit unclear. This issue is further explored and discussed in the cited paper by Forsell, Jernelöv, Blom, & Kaldö, (2022), but we have now added some clarifications of this in the introductions and a more thorough description in the Methods section under the headline now reading “Evaluation of predictive performance and clinical usefulness”:

“The benchmark **for clinical usefulness, a** balanced accuracy of 67%, was used since that **was** the balanced accuracy found in a handmade decision tree type classification algorithm that was the key part of an adaptive treatment strategy shown in a randomized controlled trial to be successful in reducing the number of failed treatments **in ICBT** (Forsell et al., 2022). This indicates that this level of predictive accuracy **was enough to** enable clinical decisions and actions, which in the end was **shown** as beneficial for the patients **whose therapists used the adaptive treatment strategy compared to those that did not** (Forsell et al., 2019). “

We also expand the text on the clinical benchmark in Limitations in the following sentences (changes highlighted):

“Another limitation is that even though an empirically established based accuracy of 67% (Forsell et al., 2022) for clinical usefulness was used, it is not necessarily generalizable over patient conditions, interventions, and clinical contexts. In that study, the clinical context, i.e. ICBT and the way it was delivered, was very similar, but the patients were treated for Insomnia. We assume an improved accuracy is always more beneficial in an adaptive treatment strategy, but it is more difficult to establish a decisive cut-off for when it is good enough and this needs to be further explored. However, our chosen benchmark is also quite similar to the cut-off of 65% previously reported by clinicians to be perceived as acceptable to act upon for clinical decisions in general (Eisenberg & Hershey, 1983).”

Forsell, E., Jernelöv, S., Blom, K., & Kaldo, V. (2022). Clinically sufficient classification accuracy and key predictors of treatment failure in a randomized controlled trial of Internet-delivered Cognitive Behavior Therapy for Insomnia. *Internet Interventions*, 100554. <https://doi.org/10.1016/j.invent.2022.100554>

In regards to the base rate of the binary outcome firstly it should first be noted that we predict a continuous outcome, which then is transformed to a binary one. The rationale for this approach has been expanded upon during previous reviews and is cited here (from “Treatment outcome ” section in the manuscript) :

“All models predicted a standardised continuous outcome value for each patient. After the continuous outcome was predicted this score was dichotomized into ‘success’ if the score was below remittance for the scale or if sufficient symptom reduction indicated treatment response, or else as ‘failure’. As such a dichotomisation of the outcome prediction was constructed after the continuous prediction had been made and a dichotomisation of the outcome itself was never conducted. ... Dichotomisation was made for the exclusive reason to facilitate comparisons to other predictive models in the field and to reflect a possible clinical guidance in line with the 67% benchmark (Forsell et al., 2022) - other continuous metrics were also calculated.”

Secondly balanced accuracy was used since it is a weighted accuracy which corrects for base rates. If the algorithms had predicted 0 or 1 for all patients, the “balanced accuracy” metric would effectively get a 50% balanced accuracy (compared to the metric of “accuracy” which might get the base rate percentage of correct).

Reviewer #4

-p. 18, l. 441: A 10% test sample seems quite small. When no k-fold- cross validation is applied, but data are split into train and test samples, a 70/30 split would be more common. I agree that large data are needed for training a good model, however, the reliability of the test depends on the sample size of the test data. Please elaborate on this point and explain the choice for a 90/10 split.

Authors:

Under the heading of “preprocessing” we designate how our “validation split” was carried out as following:

“Validation split. The datasets were split into training (90 %) and test (10 %) datasets, with training of algorithms and imputation of data based on the training datasets. Furthermore, the training dataset implemented a 10-fold cross-validation. This was done to prevent overfitting and thus get more generalisable results.”

To clarify, we have a 10% test sample, independent on the other sample. However, we also carry out a k-fold-cross validation (k = 10), which we focus on throughout the paper.

Reviewer #4

-p. 19, l. 447: Variables with more than 25% missiness were excluded. However, this rule

was not applied to some variables, including homework, comorbidities, and employment. This inconsistent approach should be explained. Furthermore, the missingness of these variables should be reported in total cases and percentages. First, this is especially important for the approach, in which these missings are statistically imputed. Second, many missings in these variables would result in a small sample size for the approach without imputation. Therefore, the sample size for each analysis should be reported to provide readers with enough information to interpret the results appropriately (maybe imputing is just better because of the larger sample size).

Authors:

We agree with the reviewer that this is a somewhat complicated and unclear aspect. We had previously written in the Methods, under the headline "Handling missing data (second primary aspect)":

"All predictor variables with more than 25 % missing values after splitting the datasets by time, treatment, data selection and validation split were excluded. Exceptions were made for variables for homework, comorbidities and employment, due to their expected predictive value"

Thus, the reason for keeping these variables even in case of high drop-out was due to their expected high predictive value. One important thing to clarify is that it was actually only one of the three variables that failed to meet the max 25% criterion - the homework variable. This was because the exception for these three variables was made before their actual missingness was investigated, and as can be seen in our addition below the variables for comorbidities and employment were below 25% in missingness.

Furthermore, this exception was only made for the imputation datasets, and not the missing removal datasets. To clarify, the missing removal datasets had missing removed after variables with 25 % missing values or more had already been excluded. That is why the variables in these datasets look different as the supplement details.

We have therefore expanded this paragraph to clarify more clearly this procedure more clearly in conjunction with adding the amount of missingness per the reviewers suggestion (changes highlighted):

“All predictor variables with more than 25 % missing values after splitting the datasets by time, treatment, data selection and validation split were excluded. Before the level of missingness was explored exceptions were made for: homework variables, comorbidities, and employment. These exceptions were made due to their expected high predictive value, and was only made in the imputation datasets, The exception was only used for the homework variables with 32-78% missing. The other two had 6% and 3% missing respectively and thus would not have been excluded even if the exception rule would not have been applied. During treatments worksheet are not mandatory to fill in, explaining the higher percentages of missing for the homework variables.”

To further clarify the sample size for each analysis is provided in full in the supplement with the full result under the heading of “Data_shape_org” where the first number designates the number of patients for that analysis.

Reviewer #4

-p. 19., l. 452: Although again there is a reference given, I’m quite sceptical about imputing the outcome variable together with its predictors. It seems obvious to me that this approach creates associations between predictors and outcome (especially in the case of a high rate of missingness; see my previous concern).

Authors:

We do understand that the reviewer has concerns about the imputing procedure, as reviewer #2 has had previously, before both revision 1 and revision 2. This issue has also been lifted during previous reviews of this manuscript in other journals, but then reviewers

with expertise in ML on the contrary encouraged a more extensive use of imputation than was originally included in the manuscript. Under Pre-processing in Methods, we explicitly cite our reasons for our imputation procedure (new additions highlighted):

“The imputation procedure also included and imputed the outcome in line with existing recommendations (Moons et al., 2006; van Ginkel et al., 2020) having empirically shown that this gave better estimates and is not negatively affected by circularity, **i.e. only confirming existing predictor-outcome relationships.** Secondly, case-removal was used, resulting in datasets with fewer patients. Finally, a sensitivity set of datasets were created for analyses where the outcome was not imputed, but predictors were.”

We thus have a sensitivity analysis where the outcome is not imputed, and as such predictors and outcome is not mixed and no risk of circularity is present, which is the concern of the reviewer. Thus, in our data we can gauge the effect of imputing outcome as well as predictors. In previous revisions we have also expanded our result section so one can view the result 1) without any imputation 2) with imputation for predictors but without outcome imputation 3) with complete imputation, and we have also previously expanded our discussion and limitation section to discuss this. We now add the below highlighted sections to further elaborate on this:

“We imputed missing data, including the outcome, and while we consider this a valid approach according to recommendations (Moons et al., 2006; van Ginkel et al., 2020) it could have influenced our scores. **The sensitivity analysis indicates that imputing predictors but not outcome decreased accuracy by 3.5% on average. We cannot know if this increase in prediction accuracy for outcome imputation was due to the procedure introducing bias despite the previous findings to the contrary (Moons et al., 2006; van Ginkel et al., 2020), or if it was because the impute-all approach were superior in a non-biased way, and it could be a combination of both.** A multiple imputation procedure could have further ensured robustness of our results **and could be a way** to quantify this uncertainty but was deemed too computationally costly due to the combinatorial increase of analyses multiple datasets would entail. However, both Moons et al., (2006) and van Ginkel et al., (2020) underscores the

preference for including outcome in imputation procedures”

Reviewer #4

-p. 2, l. 89: The second benchmark was a regression model with “fewer variables”. Please specify what “fewer” means. How many variables were in the model and how were they selected?

Authors:

Under the heading of “Variable selection” we have specified as the reviewer wishes, the passage is the following:

“‘Benchmark’ variables were a subset of the handpicked variables, consisting only of the weekly primary symptom measure, gender and age.”

The cited “fewer variables” from the reviewer comes from the introduction, with the full citation being:

“Our main aim is to evaluate the accuracy of machine learning outcome prediction, using a large sample of patients who have received ICBT in regular care, compared to a benchmark for clinical usefulness (67%; (Forsell et al., 2022) and a simpler, benchmark regression model using fewer variables.” As such we expand upon what we mean by this in the Method section.

Reviewer #4

I think that the structure (methods after discussion) is not beneficial to understand the manuscript. The results section in its current form is not clear without having read the methods. Therefore, I would recommend restructuring the manuscript. I know the structure will be given by the journal guidelines, however, in this case, maybe the authors or the editor should think about changing the structure.

Authors:

We tend to agree with this, since we also are used to another manuscript structure.

However, the current format is given by the guidelines of the journal and defer to the editor and/or copyeditor to make the decision on changing this.

REVIEWERS' COMMENTS:

Reviewer #2 (Remarks to the Author):

Everything has been addressed very well.

Reviewer #4 (Remarks to the Author):

The authors provide an improved version of their previous manuscript for which I also was a reviewer (reviewer #4). I no longer have any concerns that would prevent publication of the manuscript. I congratulate the authors on the convincing and successful revision of their manuscript.

I apologize for taking so long to respond to the last revision. I would also like to acknowledge that, despite (two?) previous rounds of review, the authors had received another comprehensive initial review with me as a new reviewer and they have responded to it very comprehensively. I'm convinced that this manuscript will make a valuable contribution to the field.

REVIEWERS' COMMENTS:

Reviewer #2 (Remarks to the Author):

Everything has been addressed very well.

Authors:

We thank the reviewer for their time and review.

Reviewer #4 (Remarks to the Author):

The authors provide an improved version of their previous manuscript for which I also was a reviewer (reviewer #4). I no longer have any concerns that would prevent publication of the manuscript. I congratulate the authors on the convincing and successful revision of their manuscript.

I apologize for taking so long to respond to the last revision. I would also like to acknowledge that, despite (two?) previous rounds of review, the authors had received another comprehensive initial review with me as a new reviewer and they have responded to it very comprehensively. I'm convinced that this manuscript will make a valuable contribution to the field.

Authors:

We thank the reviewer for their time and review.